# Sexual assault and the matrix of harm: Sexual assault survivors narrate their whole lives in more negative ways

**MacKenzie Caputo[1], Max Fineman[2], Shamus Khan[2,3] ***

1 School of Public and International Affairs, Princeton University, Princeton, New Jersey, United States of America, 2 Department of Sociology, Princeton University, Princeton, New Jersey, United States of America, 3 Department of American Studies, Princeton University, Princeton, New Jersey, United States of America

☯ These authors contributed equally to this work.
* shamuskhan@princeton.edu

**Data Availability Statement:** Interviews from the American Voices Project considered many sensitive topics, including but not limited to experiences of sexual assault. Given the breadth and detail of the interview protocol, subjects are at

## Abstract

This paper uses data from the American Voices Project, an interview study based on a random population sample, to explore the relationship between assault experiences and how people narrate their lives. Using quantitative sentiment analysis, we find that survivors of assault express significantly greater negative sentiment when asked to tell their life stories. These negative sentiments are observable throughout the entire interview, including before questions of assault are asked. Survivors of assault narrate their experiences with more anger, disgust, fear, and sadness, and less anticipation, joy, and trust than those who do not report assault experiences. We provide evidence that the negative sentiment associated with sexual violence is greater than having lost a parent and less than having experienced a significant financial hardship within the last month. We contextualize these findings with a qualitative evaluation of the interview transcripts, further substantiating our finding. Overall, this paper suggests that sexual violence is part of what we have called, drawing inspiration from the work of Beth Richie, a "matrix of harm" that structures people's lives. While our finding is consistent with what we might expect given the negative life experiences and sequalae associated with sexual assault, it has important implications. Sentimental differences in narrating life experience are an important yet relatively understudied phenomenon, and experiences of assault are rarely asked about yet may be consequential to both quantitative and qualitative accounts of social processes.

## Introduction

Sexual assault is an extremely common experience associated with significant negative sequalae: increased odds of exhibiting post-traumatic stress disorder, anxiety, depression, and suicidal ideation; there are well-documented associations between assault and physical health challenges such as chronic asthma, heart disease, and stroke [1–3]. Actuarial assessments suggest significant economic consequences—estimating the lifetime economic cost of a rape in the United States at nearly $125,000 [4]. Scholars have also demonstrated negative associations

extreme risk of being identifiable. This limits the broad public availability of data. Data from this project is available to researchers upon application through the American Voices Project, based at Stanford University. Interested scholars may be granted access to data upon successful application to their own institutional review board, and, their following protocols outlined by the American Voices Project. The data are available through request from Stanford University by emailing: americanvoicesproject@stanford.edu.

**Funding:** The authors received no specific funding for this work.

**Competing interests:** The authors have declared that no competing interests exist.

between assault experiences and future intimate relationship duration and stability as well as other social non-intimate relationships [5].

Using the American Voices Project (AVP) we provide new socially meaningful evidence of the harms associated with such violence. Using quantitative sentiment analysis, we find that those who report experiences of sexual assault narrate their lives more negatively than those who do not. When asked by an interviewer to "tell your story," those who report experiencing sexual assault narrate their lives in ways that are significantly more negative, filled with more anger, disgust, fear, and sadness, and less anticipation, joy, and trust. These negative sentiments are observable *before* interview subjects are ever asked about an experience of sexual assault. We evaluate the relative association of sexual violence with narrative sentiment and find that it is less associated with negative sentiment than having experienced significant financial hardship within the last month and more associated with negative sentiment than the loss of a parent. Inspired by Beth Richie's pioneering concept of the "matrix of violence," we suggest that assault is part of a matrix of harm that pervades the lives of survivors.

We highlight two implications of this novel finding. Sexual violence is not randomly distributed across the population; its increased prevalence among women, LGBTQIA-identifying people, and those with disabilities suggests an unequal distribution of negative sentiment on a population level. Second, major social surveys and interview projects rarely ask about experiences of sexual violence. Yet the strong association between such violence and negative sentiment suggests that a potentially meaningful determinant of life chances and experiences is omitted from both quantitative and qualitative research instruments. Scholars could be misattributing negative sentiment, or other associations, because of our failure to observe assault experiences.

## Sexual violence

We define sexual assault as sexual contact without consent or agreement [2]. This definition includes behaviors ranging from the fondling of intimate body parts to nonconsensual sexual penetration. Following work pioneered by Mary Koss [6, 7], we operationalize assault by whether or not an experience meets a behavioral definition (i.e., "have you ever experienced unwanted sexual contact?") rather than a how a subject categorizes an experience (i.e., "have you ever been sexually assaulted?"). Such an approach is standard within the literature, wherein measures of assault are behaviorally specific rather than categorical [8].

The prevalence of sexual violence is high. In the United States it is estimated that approximately 44% of women and 25% of men will experience some form of sexual violence in their lifetimes; over 20% of women will experience a completed or attempted rape [9, 10]. The strongest and most consistently observed associations of assault are prior experiences of assault [11–13] and binge drinking [14, 15]. Differences in reported rates of prevalence are often differences in measurement. Some studies estimate lifetime risk while others express current rates within the population. Some studies define sexual violence as unwanted touching, attempted rape, or rape, whereas others may include sexual harassment, or focus primarily on rape or attempted rape. Some draw upon crime victimization surveys, which have reliability concerns, and others draw upon purposive sampling. Given similar operational definitions, the rates observed by scholars have been fairly consistent for decades. AVP does not provide a definitive estimate of the rate of sexual violence, but as a national probability sample it does provide some validation of previous estimates of prevalence. Conversely, the reliability of AVP for addressing questions of sexual violence can also be evaluated by its consistency with other assault studies. We find that AVP's prevalence estimates are consistent with other studies, substantiating the overall rate of such violence and suggesting that the data can be reliably used to analyze the association between sexual violence and negative sentiment.

## The matrix of harm & the impacts of negative sentiment

We conceptualize the relationship between sexual assault experiences and negative sentiment through what we call the matrix of harm. Beth Richie's pioneering Black intersectional feminist work introduced the scholarly community to the concept of "the violence matrix" [16]. Her framework, grounded in the experiences of Black women, highlights nine modes of harm. The matrix exists along two dimensions: on one dimension she notes how states, communities, and households can be *sites* of harms. On the other, she presents the harms of physical assault, sexual assault, and emotional manipulation—harms that states, communities, and household enact in unique yet intersecting ways. Richie's concept of the "matrix of violence" emphasizes the multidimensional nature of the cultural and institutional systems that enact and perpetuate violence. Individual acts of violence are refracted through the dynamics of communities, institutions, and states to produce experiences of harm. The implication of this perspective is that both the causes and consequences of violent acts like sexual assault are shaped by their intersection with broader systems of violence.

Taking directly from Richie, we think about a *matrix of harm* reflected within the narratives of assault survivors. The concept is grounded in part in the non-causal nature of our account. Given our data and research design, we cannot claim that assault produces negative sentiments. Instead, recognizing the multiple adverse experiences associated with sexual violence, and the many negative sequalae, we conceptualize sexual violence as existing within intersecting matrices of harm, produced by individual, relational, institutional, and cultural factors. Such an approach has a firm grounding within recent ecological models of sexual violence [2]. This kind of conceptualization parallels (and is recognized within) Richie's [17] account of "how complex community dynamics, ineffective institutional responses, and broader societal forces of systemic violence intersect" to produce individual harms and influence affective and material experiences within the world.

Richie's theoretical framework "points to the utility of moving beyond quantitative studies and single-dimension qualitative analyses of the impact of violence and instead encourages designing conceptual models that consider root causes and the ways that systemic factors complicate its impact" [17]. We suggest that sentiment analysis is a way to observe matrices of harm, wherein multiple intersecting forms of marginalization, violence, and domination are experienced from the micro to the macro level. The language used by interviewees in in-depth interviews, analyzed at scale, provides a view into individual experiences of harm within this intersectional system.

We present this framework with a note of caution. The relationship between verbal sentiment and social action is relatively under-studied. While there is increasing attention to sentiments, driven in no small part by the growth of computational textual analysis, scholars do not yet have a clear account of the downstream consequences of verbal sentiment differences for individuals' lives and for broader social outcomes. The social significance of verbal sentiment has yet to be well-established empirically.

Nonetheless, within psychology—particularly in research in the psychology of emotions— the importance of sentiment is well established. In their now classic study, Schwarz and Clore [18] found that "people use their momentary affective states in making judgments of how happy and satisfied they are with their lives in general"—suggesting that feeling good or bad in a moment will impact how people report thinking about their lives in general. In the decades since, research on emotions and their impact on judgement and behavior has flourished. We cannot make a strong connection between this literature and our findings because the kind of sentiment analysis we rely upon, sentiment measured in qualitative in-depth interviews, has not been validated within the broader psychology literature as part of such emotional

assessments. The psychology of emotions nonetheless provides suggestive evidence that sentiment impacts social outcomes.

Work building off Schwarz and Clore [18] suggests caution in looking only at emotional valence (positive or negative emotion), and instead encourages scholars to examine emotion-specific phenomena (delineating, for example, between fear and anger). While we provide empirical evidence of the simple cases of positive or negative sentiment, we also provide sentiment-specific findings. Our sentiment analysis separately identifies distinct emotional states—anger, disgust, fear, sadness, anticipation, joy, and trust—all of which have strong associations with important outcomes within the psychological literature on emotions. In their review of the primary findings in the field, Lerner et al. argue that "emotion effects are neither random nor epiphenomenal. . . emotions exert causal effects on the quality of our relationships, sleep patterns, economic choices, political and policy choices, creativity, physical and mental health, and overall well-being" [19] Much more work needs to be done to connect sentiment analysis to outcomes, but if we think of significant increases in anger, disgust, fear, and sadness, and decreases in anticipation, joy, and trust as emotional manifestations, then the psychological literature suggests that these empirically observed differences are meaningful for social life.

## American Voices Project

This research draws upon the American Voices Project (AVP), a novel interview study based upon a national random population sample. From 2019 to 2021, the AVP conducted thousands of immersive interviews to provide a comprehensive image of the state of the American people (framed by its designers as a "qualitative census"). The AVP asked open-ended questions about life history, family, community, living situations, health, employment, other demographics, and sexual assault. The AVP interview transcripts not only offer information on sexual assault survivorship but also provide detail on the demographic features and life events of interviewees. Additionally, the AVP interviews offer extensive amounts of text that allows comparison between those who did and did not report experiences of assault.

To gather a nationally representative sample and guarantee adequate low-income representation in the data, the AVP used three-stage cluster sampling to identify heads of households to be interviewed (for a full account of the AVP methods, see American Voices Methodology 2021) [20]. Within the period of data gathering this paper draws upon (July 2019 through March 2020), the AVP conducted in-person recruitment, sending letters to the selected addresses, and following up with an in-person visit. During this phase, 4,208 addresses were recruited, and 1,564 interviews were conducted. At the start-date of our analysis (January 2023), 1,123 of these interviews had been transcribed and were available for analysis. In March 2020, the pandemic forced the AVP to switch to a remote interview format. During this transition, the AVP removed several highly sensitive questions from the interview script, including the question of interest in this study: the question asking about unwanted sexual contact. For this paper we primarily analyze the interviews conducted before March 2020. All analyses were re-run including the AVP sample weights. There were no notable differences. We chose to report results without sample weights because of concerns about selection into our sample.

The interviews lasted an average of 2.2 hours and included a qualitative portion and short quantitative survey at the end. The AVP designed the interviews to gather data on the state of the American public. Interviews included open-ended questions spanning the following topics: 1) life history and pivotal life events; 2) relationships with household members, family, friends, coworkers, and social support systems; 3) patterns and routines of everyday life; 4) the scope of financial hardships and reactions to such hardships; 5) primary income streams and expenses; 6) occupations of household members; 7) government program usage; 8) neighborhood,

residential history, and living situation; 9) everyday routines and community; 10) health and healthcare; 11) mental health, substance use, and emotional well-being; 12) political leaning and civic engagement; and 13) identity and sources of meaning. Protocols for the qualitative interviews included nearly 200 questions excluding built-in follow-up questions asking interviewees to elaborate on answers.

Approximately halfway into the duration of the interview, interviewers asked one question relating to sexual assault. The question listed on the interview protocol was, "Sometimes people tell us they've experienced problems due to unwanted sexual contact or uninvited sexual situations. How about for you?" While interviewers asked this question somewhat differently in practice, the selected wordings prompted interviewees to share whether they have had experiences with unwanted sexual contact.

## Methods and materials

### Data

Interviews from the American Voices Project considered many sensitive topics, including but not limited to experiences of sexual assault. Given the breadth and detail of the interview protocol, subjects who participated in the AVP are at extreme risk of being identifiable. Interview subjects were promised that "All members of our research team are legally required to keep all information provided by participants confidential and secure. We will never identify you as having participated in the study". For this and other reasons, informed consent was confirmed verbally. The extreme risk of subject identification means there are significant and important limits to the public availability of data. Data from this project has been made available to researchers upon application through the American Voices Project, based at Stanford University. Interested scholars may be granted access to data on secure servers upon successful application to their own institutional review board and by following protocols outlined by the American Voices Project. Occassionally, quotes are slightly modified, data coarsened, or not reported out of concern for confidentiality.

### Analytic strategy

We leveraged the AVP data to examine how reports of experiencing assault is reflected in broader life narration. This meant looking not simply at responses to assault questions, but to differences across the entire life-story that subjects narrated during their interview. We first constructed a binary category for each interview: "survivor of sexual assault" or "individuals who did not report experiencing sexual assault". We then undertook three analyses: (1) regression models to identify the factors most associated with survivors of sexual assault; (2) computational text analysis to compare the sentiments and language of survivors' interviews compared to those who did not report an assault experience; (3) qualitative analysis of transcripts to further explore sexual assault-related patterns in the data. The first analysis was a validity check of the AVP, ensuring that core findings of the broader sexual assault literature were similar to this relatively new, untested data source. The goal of the second and third analyses was to better understand how assault experiences were associated with life narratives, in general.

### Coding sexual assault

To create a sexual assault variable, we coded the answer to the "unwanted sexual contact" question in the AVP. The AVP interview protocol included about 200 questions, and the sexual assault question was not asked in every interview; the 382 interviews that did not ask this

question were discarded from the analysis. We cannot know how or why interviewers decided not to ask this question, and it may be a source of bias in our findings. Some respondents (37 interviewees) divulged their survivor status without an interviewer inquiring, therefore we included these in our sample. After the pandemic began, the interview protocol no longer asked about unwanted sexual contact. Yet 10 interviewers nonetheless still asked this question (and subjects chose to answer); these are included in the analysis of this paper.

Some interviewees' responses to the "unwanted sexual contact" question highlighted challenges to our binary sorting. For example, a few interviewees who responded in the negative to the question said, "I've never been sexually assaulted if that's what you mean". However, several of these interviewees then narrated an experience that met the operational definition of sexual assault. This is consistent with broader findings in the literature about resistance to categorizing certain experiences as assault [8]. Using a behavioral operationalization of sexual assault, rather than a measure based on interviewees own categorization of their experience, we coded such interviewees as having experienced sexual assault. A second binary sorting challenge arose when some interviewees directly refused to answer the "unwanted sexual contact" question. Third, some interviewees responded to the inquiry but failed to answer the question. Such respondents discussed their own consensual experiences, nonconsensual experiences of friends or family, or positions on controversial issues such as abortion or marriage equality. Furthermore, some interviewees misinterpreted the question, discussing their experience sexually assaulting another person or being accused of such. Together, 37 interviews were removed from the data set because interviewees refused to answer the "unwanted sexual contact" question or because the transcripts failed to clearly reveal the status of survivorship. Where logically possible, analyses reported in this paper were re-run to evaluate all inclusion criteria. None of the decisions to remove/include interviews impacted the reported findings. The main analysis in this paper uses a sample of 458 interviews from the AVP. Some analyses use slightly fewer than 458 interviews because covariates and/or gender variables are missing (46 instances), or predictor variables are missing (53 instances), or gender variables are missing (6 instances). Table 1 contains information on the samples used for different analyses.

We sought to assess why questions about assault experiences were asked in some interviews and not others. Most interviewers asked this question sometimes but not all the time: 12% of interviewers never asked the assault question, 7% always asked, and 80% asked sometimes but not all the time. If we compare the demographics of those asked with those not asked there are no notable differences. We cannot rule out that interviewers are selecting who to ask the assault question based on their interactions with subjects. Yet with so many questions (just under 200) in a two-hour interview, the sexual assault question is not an outlier in being asked irregularly. Selection into being asked the assault question is, nonetheless, an unobservable dynamic that may be biasing our results.

**Table 1. Cases used in analyses.**

| | Logistic Regression & summary statistics | Sentiment Analysis and Weighted Log Odds Model |
|---|---|---|
| Interviewees Asked About Assault | 456 | 456 |
| Interviewees failed to answer assault question, missing gender info, or answered gender question outside binary answer | 77 | 43 |
| Interviewees Volunteered Survivor Status (w/o being asked) | 39 | 39 |
| Total number of interviewees | 418 | 452 |

## Regression analysis

We employed two logistic regressions to examine the relationships between varying demographic variables and sexual assault survivor status. These variables (identified by the literature on sexual violence as important factors) included gender, ethnicity, age, marital status, educational attainment, political leaning, income, and geography. The AVP did not ask about sexual orientation. Minimal, non-patterned missingness appeared across these variables, and interviewees identifying as gender non conforming were removed due to lack of substantial data, resulting in the combined removal of 41 interviews (N = 418). See Table 2 for summary statistics on age, educational attainment, income bracket, marital status, and political leaning. To analyze the relationships between these demographic variables and sexual assault experiences, we conducted two logistic regressions. The first model utilized the basic demographic variables provided in the AVP quantitative data set, including gender, age, marital status, race, level of education, income, and political orientation. The second model added state-level fixed effects (strongly associated with political variation). These regressions were primarily used to evaluate consistency between AVP and other studies of assault.

## Computational text (sentiment) analysis

Responses to the question on sexual violence itself were, on average, limited to just a few words (median word count = 37); as a result, qualitative interpretation of these narratives is limited. We therefore used sentiment analysis and a weighted log odds model to examine how the language of the interviews overall differ between those who did and did not report experiencing assault. To construct the corpus of text we analyzed, all punctuation was removed, as were all interviewer questions or comments. Stop words were also removed (i.e., "the," "is," and "or"). The AVP used a number of placeholders to replace personally identifiable information in the transcript text. We used these placeholders, various names left in transcripts, and proper nouns that nominated one or two interviews but failed to appear elsewhere in the data to create an AVP-specific stop word list. We removed all AVP-specific stop words from the data.

We consider the text of each interview as a combination of individual words and the overall sentiment of each interview as the sum of the sentiment content of the words. The sentiment analysis used two different sentiment lexicons: the Bing lexicon and the NRC lexicon. Both lexicons use unigrams (single words) to measure sentiment. The Bing lexicon is a standard sentiment lexicon that categorizes words as either positive or negative (thought of as "valence" in the psychology literature on emotions). To provide greater validity for our findings, and, to look at emotional responses beyond valence, we also use the NRC lexicon. The NRC lexicon categorizes words as positive or negative but additionally sorts words into eight different emotions: anger, anticipation, disgust, fear, joy, sadness, surprise, and trust. In the NRC lexicon, a given word can be assigned multiple emotions. For example, NRC assigns the word "abandon" to both fear and sadness. Neither of these lexicons assign all words to a sentiment or emotion category.

While sentiment lexicons enable the quantification of text, they have limitations. Given that these lexicons focus on unigrams, they can misrepresent sentences or phrases. For example, the phrase "not good" would receive a non-negative tally despite expressing a negative sentiment. Despite such shortcomings, scholars argue that sentiment lexicons remain useful tools for social analysis [21]. Our qualitative analysis helps us evaluate the potential extent of such misrepresentations; it suggests that our estimates are likely conservative.

The NRC lexicon was developed using Mechanical Turk [22]. The Bing lexicon was trained in 2004 using consumer reviews. Both Bing and NRC are widely used lexicons for sentiment

**Table 2. Summary statistics.**

| | | N | Assault reported | Assault not reported | % Assault reported |
|---|---|---|---|---|---|
| Ungrouped | | | | | |
| | Total | 418 | 153 | 265 | 37% |
| Gender | | | | | |
| | Man | 141 | 24 | 117 | 17% |
| | Woman | 277 | 129 | 148 | 47% |
| Race | | | | | |
| | White | 213 | 88 | 125 | 41% |
| | Black | 84 | 23 | 61 | 27% |
| | Hispanic or Latino | 88 | 27 | 61 | 31% |
| Age | | | | | |
| | 18–24 | 47 | 19 | 28 | 40% |
| | 25–34 | 101 | 48 | 53 | 48% |
| | 35–44 | 67 | 20 | 47 | 30% |
| | 45–54 | 58 | 23 | 35 | 40% |
| | 55–64 | 75 | 22 | 53 | 29% |
| | 65+ | 70 | 21 | 49 | 30% |
| Marital Status | | | | | |
| | Cohabitation | 39 | 15 | 24 | 38% |
| | Divorced | 56 | 20 | 36 | 36% |
| | Married | 122 | 39 | 83 | 32% |
| | Separated | 29 | ** | ** | ** |
| | Single (never married) | 147 | 61 | 86 | 42% |
| | Widowed | 25 | ** | ** | ** |
| Income ($) | | | | | |
| | < = 12,000 | 90 | 32 | 58 | 36% |
| | 12,001–24,000 | 70 | 26 | 44 | 37% |
| | 24,001–36,000 | 46 | 23 | 23 | 50% |
| | 36,001–48,000 | 48 | 17 | 31 | 35% |
| | 48,001–72,000 | 43 | 17 | 26 | 40% |
| | 72,001->120,000 | 79 | 22 | 57 | 28% |
| Education | | | | | |
| | Less than HS | 49 | 13 | 36 | 27% |
| | High School | 115 | 42 | 73 | 37% |
| | Some college | 136 | 59 | 77 | 43% |
| | Bachelor's or higher | 118 | 39 | 79 | 33% |
| Political Leaning | | | | | |
| | Strong Democrat | 94 | 41 | 53 | 44% |
| | Democrat | 59 | 21 | 38 | 36% |
| | Lean Democrat | 56 | 21 | 35 | 38% |
| | Independent | 113 | 42 | 71 | 37% |
| | Lean Republican | 26 | ** | ** | ** |
| | Republican | 22 | ** | ** | ** |
| | Strong Republican | 41 | 12 | 29 | 29% |

Groups/percentages with "**" are not reported for confidentiality.

analysis but are not well established within the literature for narrative life history; our findings should be interpreted with that in mind.

We merged the Bing lexicons with the cleaned AVP text data, counted the total number of positive words and the total number of negative words in each interview, and converted them to percentages. The percentage of positive words in each interview measures the share of all non-neutral words used in a given interview that the Bing Lexicon categorized as positive. The percentage of negative words then corresponds to the complement percentage of the positive word percentage. These percentages measure the total positive and negative sentiment of each transcript. We analyzed these sentiment percentages between those who did and did not report experiences of sexual assault and re-ran these analyses by gender. The same sentiment measurement approach was used with the NRC lexicon.

To understand the relative magnitude of the sentiment results for our sexual violence variable, we conducted a Test of Comparative Impact, running the same sentiment analysis methodology on two additional binary variables. These variables represented hardships different from sexual assault: financial hardship and parental death. We chose to run these analyses with the study population (those interviews that asked a sexual assault question and where an answer to this question was clear). We chose to use the study population because it allows our estimates to be directly comparable to our assault estimates. This first comparative hardship variable indicated whether interviewees received a housing voucher from the government in the last month. The second comparative hardship variable corresponded to whether interviewees lost one or more parents. Sentiment analysis with these hardship variables provide a metric of relative association against which we can compare the magnitude of the influence of sexual assault on sentiment. The 'housing voucher' variable offered a comparison between the sentiments associated with financial hardship and those associated with sexual assault but included a time constraint absent from the sexual assault variable. The variable for financial hardship indicated the receipt of a housing voucher in the preceding month (5.4% of subjects responded they had received a voucher in the last month). Comparing recent financial hardship to sexual assault sets a high bar for comparison, since most sexual assaults would have happened months or likely years earlier. Thus, we included the second comparative impact variable—the death of one or more parents (54.2% of interview subjects reported this experience). Death of a parent provides a useful baseline against which to compare the magnitude of the sexual assault association as it measures a common life event that can occur over a similarly broad time scale.

### Weighted log odds model & qualitative examination

To explore what aspects of interviewees' language explain the results of the sentiment analysis, we employed a weighted log odds model. This model measures how the usage or frequency of a given feature, such as a word, varies across groups of documents. In our case, the weighted log odds model estimates differences in word usage frequency between those who did and did not report a sexual assault experience. These estimates help us identify words that are much more prominent in one group than the other. A naive log odds method alone fails to account for variability in the frequency of different words in the data. Because the total number of times each word is used in the entire corpus varies by word, the resulting log odds differences misrepresent feature importance, overestimating the importance of words that are just used more by everyone. The weighted log odds model controls for this variability in word frequency. As a result of this standardization, the log odds are comparable between words. A positive log odds indicates stronger tendency among survivors to use the word and negative log odds indicates the opposite. The weighted log odds model utilizes word counts from a background corpus that ultimately facilitates the detection of differences even in very frequently

used words—a mechanism with which other word frequency methods have struggled. We split the data according to survivor status. We then deployed the "tidylo" package to model the weighted log odds of the words within each corpus [23]. We further split the data according to gender and re-ran the weighted log odds model.

Finally, while the weighted log odds model detected words that systematically separated the interviews of survivors from those who did not report an experience of assault, this model could not reveal the broader context of each word's usage. To further investigate the words and sentiments specific to survivors' interviews, we qualitatively investigated how the words detected by the weighted log odds model appeared in the AVP interviews. This involved reading the portions of interview transcripts where such words were used, examining the context surrounding the usage of each word, and providing interpretive analysis of this context.

## Results and discussion

The AVP data used in this paper shows that over 36% of those interviewed reported at least one experience of sexual assault. Of those who experienced assault, 66% identified as women. Overall, as shown in Table 1, 17% of men and 46% of women reported assault experiences. Sexual violence researchers have consistently found a similar rate of violence. The AVP showed these patterns to be racialized: Asian, Black, and Hispanic/Latino interviewees reported the lowest rates of assault while white respondents (white women in particular) and Multiracial respondents reported higher rates of assault. We also found also extremely high rates for American Indians, but the small number of observations makes these estimates less precise. This result is consistent with other reported racial/ethnic differences in assault experiences [24, 25], though the findings about racial/ethnic differences in assault are more mixed within the broader literature.

### Regression results

Table 3 reports the estimated associations between a series of individual-level variables and sexual assault across the logistic regressions. The first model included gender, age, marital status, race, education, political leaning, and income as explanatory variables. As expected, gender demonstrated a statistically significant association with sexual assault survivorship across both models. Comparing by gender, women have, on average, 7.4–8.0-times higher odds of reporting sexual assault than men, depending on the model. Further, White interviewees have, on average, 3.5–3.9 times higher odds of reporting sexual assault than Black interviewees and 2.4–2.8 times higher odds than Hispanic/Latino interviewees. Overall, the prevalence findings and regression analysis are roughly consistent with the broader literature on sexual violence. These results provide support for using the AVP to understand a wider range of associations between sexual violence and other life experiences. The results are robust to the inclusion of state fixed effects, suggesting that time-invariant differences between states cannot explain the observed differences. These findings use a national probability sample to support the prevalence rates reported in the literature.

### Sentiment and language: Comparing survivors to those who did not report an experience of assault

Our sentiment analysis considered the text of each interview as a combination of individual words. Figs 1 and 2 display the proportion of positive sentiments expressed in the AVP interviews according to the NRC and Bing lexicons. These figures depict the proportions of non-neutral words in each interview assigned to the "positive" category. The proportions of

**Table 3. Logistic models examining variables associated with survivor status.**

| | Dependent Variable: Assault Reported | |
|---|---|---|
| | **(1)** | **(2)** |
| Gender: Woman | 2.00*** (0.32) | 2.08*** (0.37) |
| Age: 25–34 | 0.39 (0.44) | 0.77 (0.52) |
| Age: 35–44 | -0.51 (0.52) | -0.03 (0.63) |
| Age: 45–54 | 0.10 (0.52) | 0.68 (0.63) |
| Age: 55–64 | -0.53 (0.49) | -0.30 (0.60) |
| Age: 65+ | -0.16 (0.53) | 0.62 (0.66) |
| Marital Status: Divorced | -0.21 (0.53) | -0.98 (0.61) |
| Marital Status: Married | -0.42 (0.47) | -0.85 (0.54) |
| Marital Status: Separated | -0.61 (0.65) | -0.69 (0.73) |
| Marital Status: Single (never married) | -0.05 (0.45) | -0.38 (0.52) |
| Marital Status: Widowed | -0.38 (0.65) | -0.73 (0.77) |
| Race: Asian | -0.94 (0.97) | -1.24 (1.18) |
| Race: Black | -1.26*** (0.36) | -1.37*** (0.49) |
| Race: Hispanic or Latino | -0.88*** (0.34) | -1.02** (0.44) |
| Race: Multicultural | -0.81 (0.59) | 1.01 (0.68) |
| Race: American Indian | 1.71 (1.22) | 1.18 (1.44) |
| Race: Pacific Islander | -12.72 (888.74) | -14.31 (2399.54) |
| Education: High School | 0.31 (0.44) | 0.56 (0.53) |
| Education: Some college | 0.63 (0.44) | 1.12** (0.53) |
| Education: Bachelor's or higher | 0.15 (0.51) | 0.50 (0.59) |
| Income ($): 12,001–24,000 | 0.16 (0.39) | -0.15 (0.45) |
| Income ($): 24,001–36,000 | 0.76* (0.44) | 0.60 (0.52) |
| Income ($): 36,001–48,000 | -0.25 (0.44) | -0.70 (0.50) |
| Income ($): 48,001–72,000 | 0.54 (0.48) | 0.25 (0.54) |
| Income ($): 72,001–120,000 | 0.32 (0.47) | 0.08 (0.54) |
| Income ($): >120,000 | -0.66 (0.63) | -1.04 (0.75) |
| Politics: Democrat | -0.82** (0.41) | -0.62 (0.47) |
| Politics: Lean Democrat | -0.53 (0.41) | -0.64 (0.48) |
| Politics: Independent | -0.33 (0.36) | -0.24 (0.42) |
| Politics: Lean Republican | 0.48 (0.57) | 0.74 (0.64) |
| Politics: Republican | -1.07* (0.64) | -0.88 (0.80) |
| Politics: Strong Republican | -1.01** (0.48) | -0.55 (0.54) |
| State: State A | | -1.81* (1.06) |
| State: State B | | -1.90* (1.00) |
| State: State C | | -1.88* (1.07) |
| State: State D | | -2.24* (1.18) |
| Constant | -1.35* (0.74) | -0.99 (1.17) |
| Observations | 411 | 411 |
| Log Liklihood | -221.02 | -191.49 |
| Akaike Inf Crit. | 512.04 | 532.98 |

*p<0.1;

**p<0.05;

***p<0.01

Baselines: Gender = Male; Age = 18–24; Marital Status = Cohabiting; Race = White; Education = Less than High school; Income = Less than 12,000; Politics = Strong Democrat

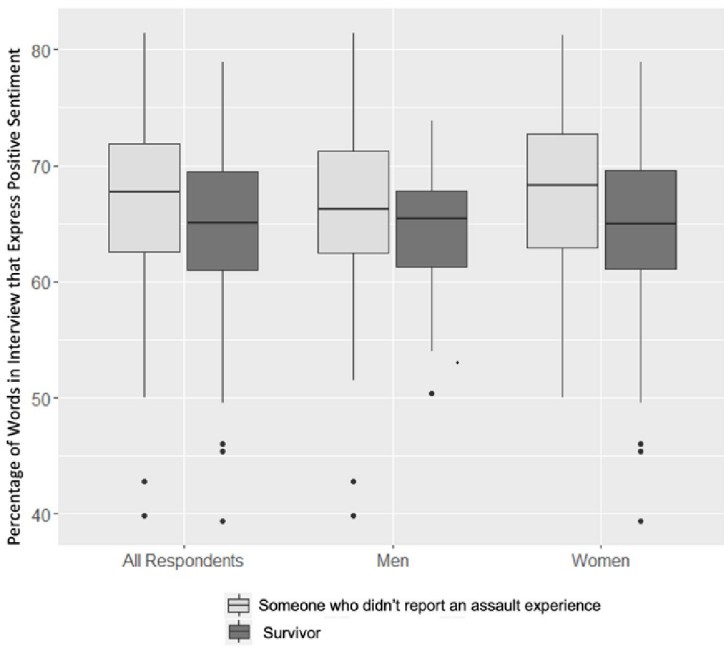

**Fig 1. Positive interview sentiment by survivor status and gender (NRC lexicon).**

negative words are excluded from the figures as these proportions are simply the complement of the positive proportions. These figures allow for the comparison between genders.

The NRC lexicon and the Bing lexicon produced similar sentiment assignments for the AVP interviews. While the median percentage of positive sentiment words among those who

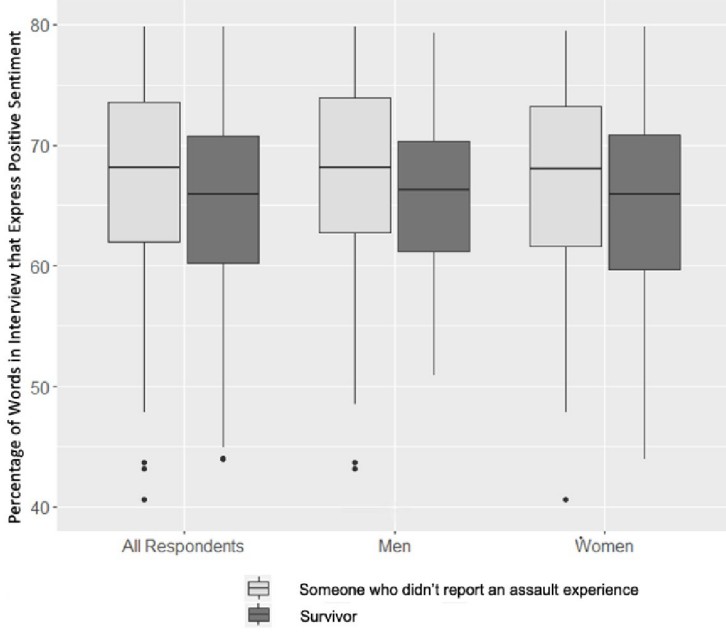

**Fig 2. Positive interview sentiment by survivor status and gender (Bing lexicon).**

Table 4. T-tests of positive sentiment between those who did and did not report assault experience.

| | | Reported Assault | Did Not Report Assault | P-Value |
|---|---|---|---|---|
| NRC | | | | |
| | All Respondents | 64.9 | 67.2 | 0.001 |
| | Men | 64.6 | 66.2 | 0.139 |
| | Women | 65.0 | 67.7 | 0.001 |
| Bing | | | | |
| | All Respondents | 65.7 | 68.4 | 0.003 |
| | Men | 66.0 | 68.2 | 0.268 |
| | Women | 65.7 | 68.5 | 0.007 |

did not report experiences of assault was around 68%, the same percentage among those who did report assault experiences was around 65%. These values remained relatively constant among the men and women in both figures. In addition to the median, the interquartile range (IQR) reflected a similar shift between survivors and those who did not report experiences of assault across the lexicons and genders. However, while the NRC lexicon found a downward shift in the IQR between those men who did not and those who did report experiences of assault, the median proportions of positive sentimental words in these two classes of men appeared almost constant. Here, the Bing lexicon deviated from the NRC lexicon, showcasing a noticeable decrease in the proportion of positive words in men's interviews. When compared to the NRC lexicon, the Bing lexicon produced larger IQRs among all the categories. The Bing Lexicon contains more positive/negative words than the NRC lexicon contains, which may explain the larger IQRs above. Both lexicons indicate that survivors' interviews included greater proportions of negative sentiments than those who did not report experiences of assault. Table 4 reveals the mean proportion of positive words among survivors and those who did not report experiences of assault across genders and lexicons, with higher scores meaning more positive overall sentiments. The differences in means mirrored the differences in medians displayed in the Figs 1 & 2.

The finding of increased negative sentiment among survivors is expected given that survivors disclosed experiences of sexual assault—an event that is likely to result in answers filled with negative sentiment. However, the AVP interview protocols included nearly 200 questions, only one of which asked about unwanted sexual contact. The mean and median number of words that survivors used to answer the question was 61 words and 37 words, respectively. These few words pertaining to sexual assault, experiences survivors refrained from describing in depth, do not account for the greater degrees of negative emotions expressed by survivors. We removed two negatively charged words that survivors frequently used to disclose assault from the corpus of interviews: "rape" and "raped". The findings did not change (see Table 5).

To further interrogate our result, we exploited the temporality of the interviews. In the interviews that included a sexual assault question, that question was asked approximately halfway into the duration of most interviews. Table 6 reports the results a sentiment analysis run

Table 5. Robustness check: T-tests of positive sentiment between those who did and did not report assault experience, removing "rape" and "raped" (Bing lexicon).

| | Reported Assault | Did Not Report Assault | P-Value |
|---|---|---|---|
| All Respondents | 65.8 | 68.4 | 0.004 |
| Men | 66.0 | 68.3 | 0.267 |
| Women | 65.7 | 68.5 | 0.009 |

**Table 6. Robustness check: Regression comparing positive sentiment between those who did and did not report assault experience before assault question asked (Bing lexicon).**

|  | Dependent Variable: Positive Sentiment |
| --- | --- |
|  | Values |
| Assault | -0.03** (0.01) |
| Constant | 0.64*** (0.05) |
| Observations | 324 |
| R-squared | 0.27 |
| Adjusted R-Squared | 0.21 |
| Residual Standard Error | 0.09 (df = 299) |
| F Statistic | 4.51*** (df = 24; 299) |

*p<0.1;

**p<0.05;

***p<0.01

for only the portion of interview text before the sexual assault question was ever asked; again, the results remain the same. This indicates that asking about a negative experience (sexual assault) is unlikely to be driving the differences in sentiment between these interviews. We say "unlikely" because a concern about priming remains. While the informed consent process did not prime respondents to think about sexual assault as it did not mention this as a discussion topic directly, we cannot rule out that subjects began to think about their assault experiences during the informed consent process wherein they were made aware that they would be asked to talk about meaningful experiences in their lives.

To understand the relative magnitude of the sentiment results, we conducted a Test of Comparative Impact, running sentiment analysis (with both the Bing and the NRC lexicons) on two additional binary variables. These variables represented hardships different from sexual assault: financial hardship (operationalized as receipt of a housing voucher in last month) and parental death. Sentiment analysis with these hardship variables provided a metric of relative impact to determine the magnitude of sentiment difference between those who did and did not report experiences of assault. As previously noted, while the 'housing voucher' variable offered a comparison between the sentiments associated with financial hardship and those associated with sexual assault, this variable involved a time constraint absent from the sexual assault variable. Consequently, comparing recent financial hardship to lifetime experience of sexual assault sets a high bar for comparison because recent adverse events can be expected to weigh more heavily on sentiment in an in-depth interview than events from an individual's distant past. Nonetheless, if the magnitude of the sentiment difference between survivors and those who did not report an experience of assault is comparable to that for recent financial hardship, it would suggest that the relationship between sexual assault experiences and sentiment is quite strong. We also included a second comparative impact variable which is measured on a similarly broad temporal scale—the death of one or more parents.

Fig 3 and Table 7 indicate that each hardship—sexual assault, death of parent(s), and recent financial hardship—corresponded to a decrease in the average expression of positive sentiment. This provides face validity to our sentiment analysis. The extent to which these decreases in positive sentiment occurred varied by hardship. Assault survivors expressed positive sentiment, on average, nearly 3% less than those who did not report an experience of assault. This value dropped to just above 2% when comparing interviewees with living parents to those with one or more deceased parents. The recent financial hardship group demonstrated the largest

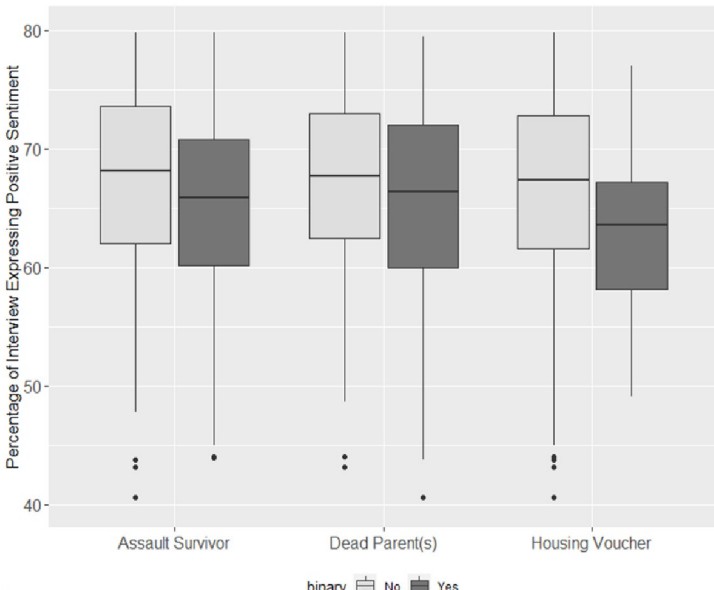

**Fig 3. Comparative impact-positive interview sentiment by hardship (Bing lexicon).**

difference in average positive sentiment with interviewees who did not receive a housing voucher in the previous month expressing, on average, over 4.5% greater positive sentiment than those who recently received a government housing voucher. Sexual assault corresponded to a greater difference in positive sentiment than that of parental death(s) and a lesser difference in sentiment than that of immediate severe financial hardship. Replicated analysis using the NRC lexicon provided equivalent results.

To further explore interview sentiments, and to do so in emotionally specific ways, we used the NRC lexicon to identify eight emotions within the interviews: anger, anticipation, disgust, fear, joy, sadness, surprise, and trust. Fig 4 compares the percentage of words in survivors' interviews expressing these emotions to that of those who did not report an assault experience. Table 8 presents the results of t-tests that examined the statistical significance of the differences produced in Fig 4.

We find that survivors narrate their experiences with more anger, disgust, fear, and sadness, and less anticipation, joy, and trust. Again, these results are consistent if the narratives are restricted to answers provided before the assault question was ever asked. All positively coded emotions are lower for survivors, and negatively coded ones are higher. Across two lexicons and two sentiment scales, survivors consistently expressed greater degrees of negative emotions compared to interview subjects who did not report an experience of assault. Robustness checks suggest that these findings are not driven by respondents narrating experiences of assault, nor by being negatively impacted in an interview by having been asked about an

**Table 7. Comparative impact—T-test comparing positive interview sentiment by hardship (Bing lexicon).**

|  | Yes | No | P-Value |
|---|---|---|---|
| Reported Assault | 65.7 | 68.4 | 0.003 |
| Dead Parent(s) | 66.4 | 68.4 | 0.019 |
| Housing Voucher | 63.1 | 67.7 | 0.020 |

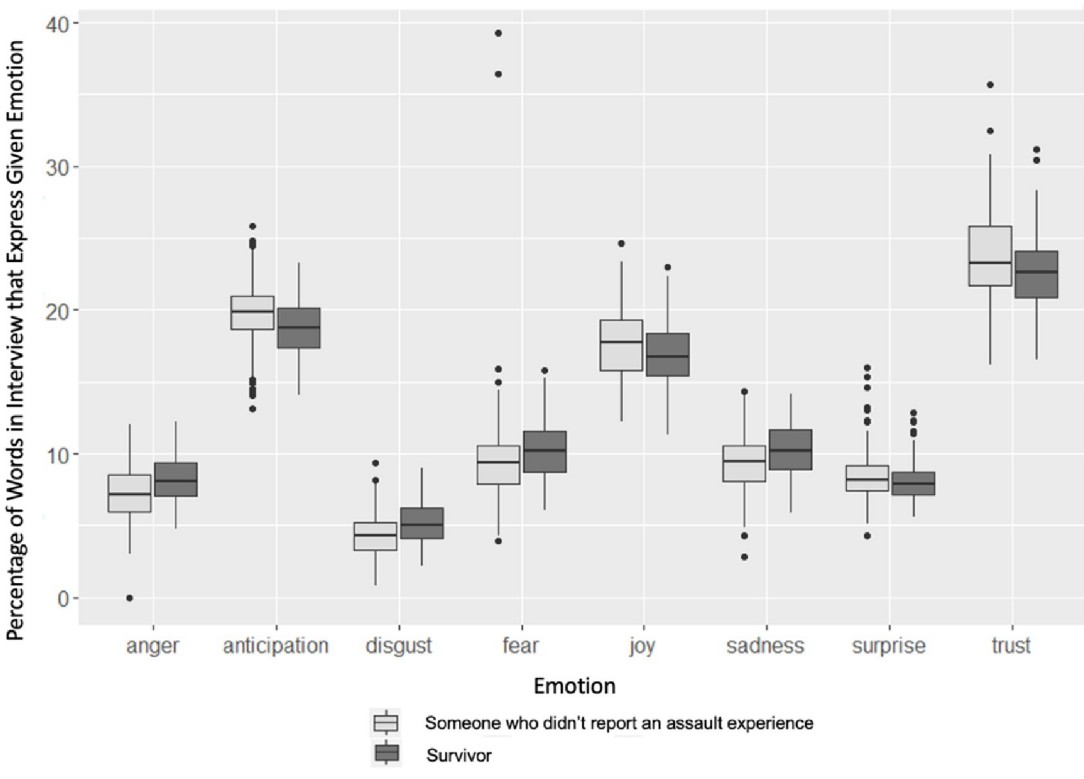

**Fig 4. Percentage of interviews spent expressing emotion by survivor status.**

experience of assault. However, it is, again, worth noting we cannot rule out some effects of priming because of the informed consent process.

## Weighted log odds model

Our weighted log odds model identified words that systematically separated the two categories of interest by creating a probabilistic model of word use that accounted for the different frequencies with which words were used. Unlike other word frequency models, weighted log odds models use regularization, which facilitate the detection of differences between those who did and did not report an assault experience even in frequently used words. A positive log odds score indicates a stronger tendency of a given group to use a word; negative log odds

**Table 8. T-test for each emotion comparing those who did and did not report assault experience.**

| Emotions | Reported Assault | Did Not Report Assault | P-Value |
|---|---|---|---|
| Anger | 8.0 | 6.9 | 0.000 |
| Anticipation | 19.0 | 20.0 | 0.000 |
| Disgust | 5.1 | 4.2 | 0.000 |
| Fear | 10.1 | 9.4 | 0.035 |
| Joy | 16.9 | 18.1 | 0.000 |
| Sadness | 10.3 | 9.4 | 0.000 |
| Surprise | 8.1 | 8.3 | 0.108 |
| Trust | 22.6 | 23.6 | 0.003 |

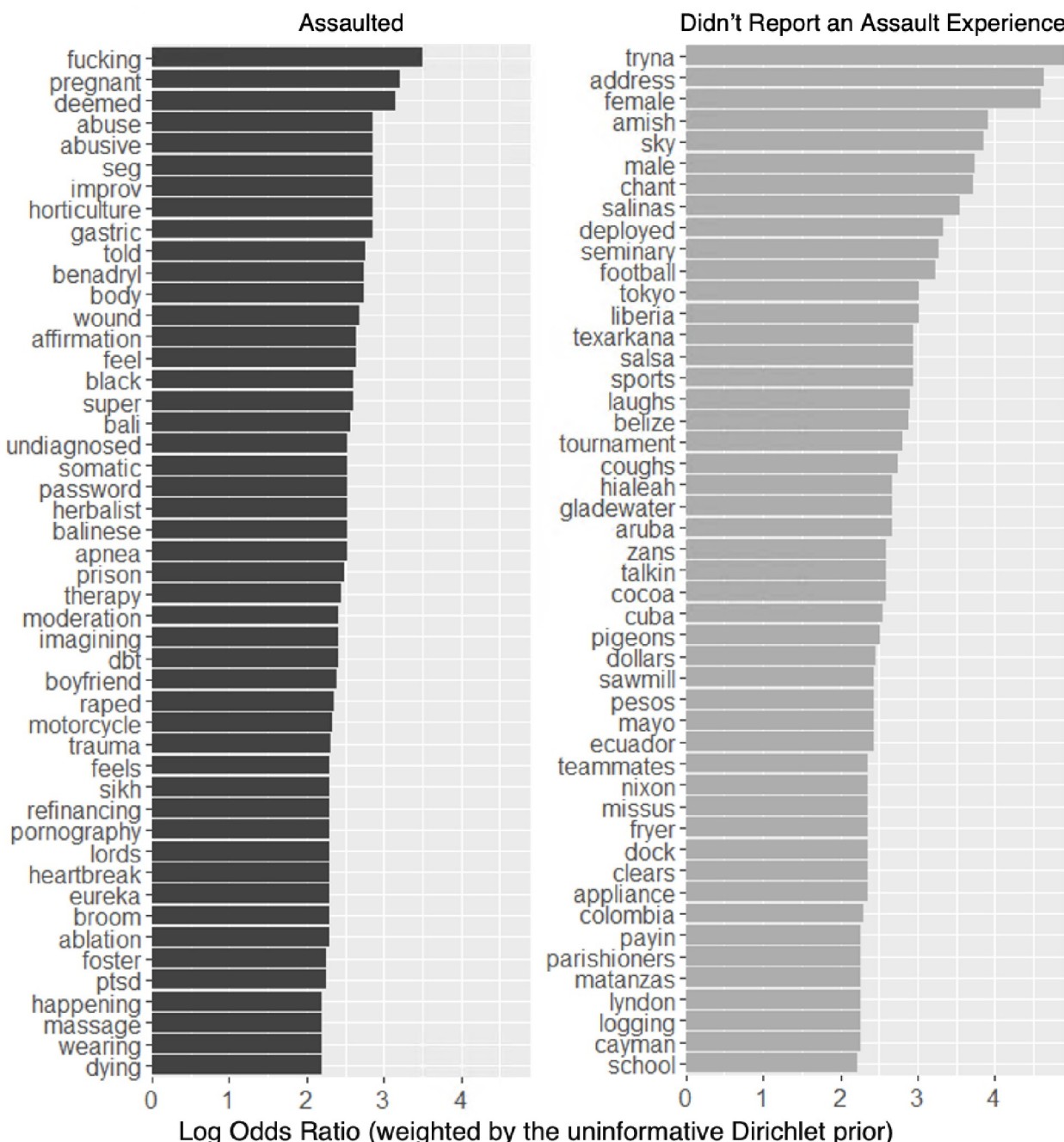

**Fig 5. Words specific to survivor's interviews vs words specific to the interviews of those who didn't report assault.**

score indicates the opposite. Fig 5 illustrates the 50 words with the greatest weighted log odds score for the two classes of interviews: those where the interviewee reported an assault and those where the interviewee did not.

For survivors, the top five words with the highest weighted log odds ratio were "fucking," "pregnant," "deemed," "abuse," and "abusive". Despite this mass of text and breadth of topic of these interviews, the words most specific to survivors shared remarkable adjacency to sexual

assault. Three of these words carry a definite negative sentiment ("fucking," "abuse," and "abusive") while the remaining two words among the five most-used are ambiguous. The word "fucking" is not only a synonym for "sex" but connotes an anger reflective of the sentiment analysis results in which survivors demonstrated significantly higher levels of anger throughout their interviews than those who did not report an experience of assault. The word "pregnant" is likely to be driven largely by gender: women exhibit higher likelihoods of survivorship and are more likely in general to use this word. The fourth and fifth most specific words to survivors' interviews, "abuse" and "abusive," are clear negative sentiments. While the weighted log odds model identified the words most specific to survivors' interviews, this model did not control for the number of survivors that used a given word, merely the number of times a word was mentioned within the corpus of survivors' interviews. As a result, the model could have assigned some words high weighted log odds scores (because a few survivors frequently used those words) despite such words only appearing in a few interviews. To account for this limitation, Fig 6 compares the 50 words with the highest weighted log odds scores (for survivors) to the number of survivors that used each word. Fig 7 visualizes the same comparison but separates on gender. The latter depicts the 30 words most specific to female survivors and the 30 words most specific to male survivors against the number of respective female or male survivors that used each word.

In Figs 6 and 7, the vertical axes reveal the specificity of a given word to the corpus of survivors' interviews. All words sit relatively high on the vertical axis because only the words with the highest weighted log odds for survivors were included in the plots. The horizontal axes reveal the number of survivors that used a given word. For visual purposes, these axes are skewed (horizontal axes do not depict even intervals). Given the quantity of data in Fig 7, this

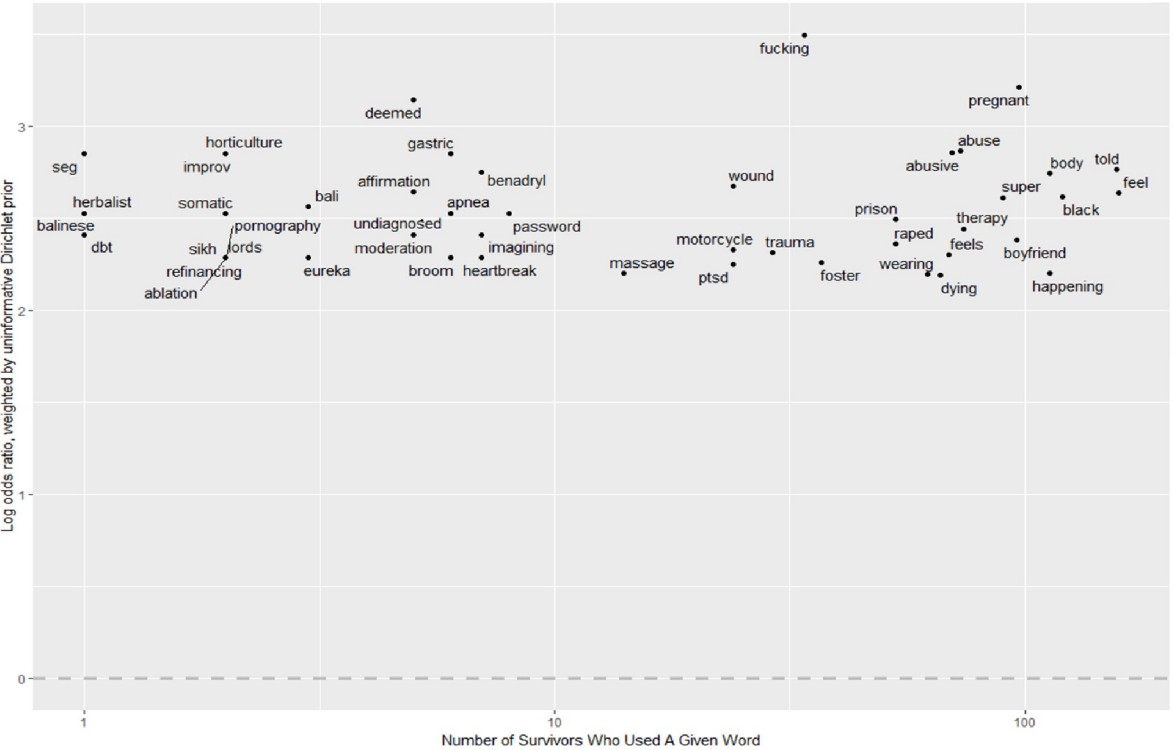

**Fig 6. Top 50 words most specific to survivors' interviews against number of survivors who mentioned word.**

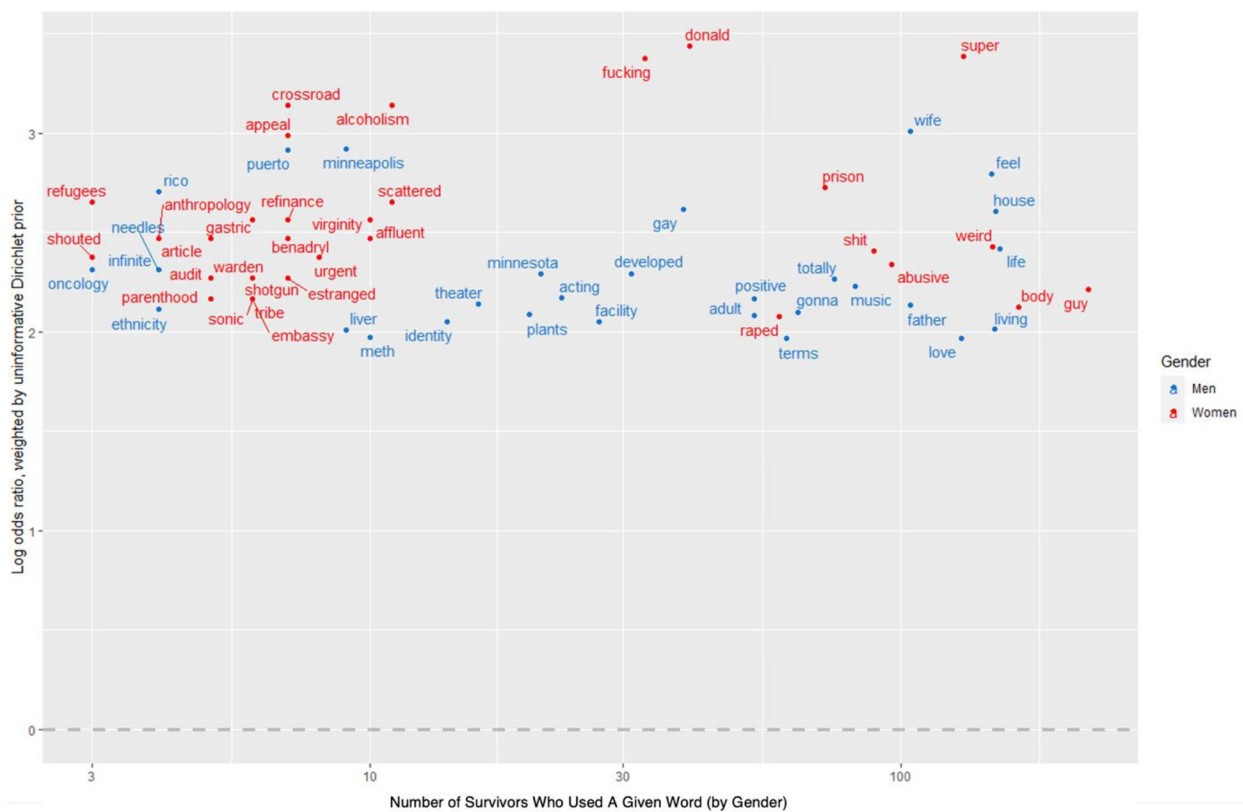

**Fig 7. Words most specific to survivor's interviews against number of survivors who mentioned word (by gender).**

figure excludes words that appeared in fewer than three interviews, and thus has a slightly different horizontal axis than that of Fig 6.

Words that appear in the upper right quadrant of Figs 6 and 7 demonstrate specificity and consistency. The weighted log odds model not only identified these words as specific to survivors' interviews, but the model derived this specificity from many interviews, indicating a pattern among survivors. Fig 6 reveals that four of the five words with the highest weighted log odds score were used by more than 35 survivors. Thus, "fucking," "pregnant," "abuse," and "abusive" were highly specific words to many survivors' interviews. Importantly, several other words in this top right quartile also share a remarkable adjacency to sexual assault. These words include "raped," "trauma," "therapy," and "PTSD". When breaking down word specificity by gender, as seen in Fig 7, "shit" and "guy" emerged as additional words that helped the model separate female survivors from those women who did not report an assault experience. Furthermore, "alcoholism" and "virginity," while not used by as many female survivors as some other words with high weighted log odds scores, demonstrate a negative adjacency to sexual assault. For male survivors, fewer words that typically circulate sexual assault literature appeared as specific to many male survivors' interviews. However, one word, "gay," scored high on the weighted log odds scale and appeared in over 15 interviews with male survivors. The appearance of this word is not fully interpretable, as the AVP did not ask about sexuality, but could suggest support for current research findings that gay men are at higher risk of sexual assault than heterosexual men. In Fig 7 the frequency of the world "Donald" does not reference a respondent or their partner, but instead represents "Donald Trump".

**Table 9. Use of assault specific language by those who did and did not report an assault.**

|  |  | Average Use per Interview | Number of Interviews |
|---|---|---|---|
| Reported Assault |  |  |  |
|  | Abuse | 2.66 | 73 |
|  | Abusive | 2.37 | 70 |
|  | Raped | 1.66 | 53 |
| Did not Report Assault |  |  |  |
|  | Abuse | 1.29 | 68 |
|  | Abusive | 1.28 | 56 |
|  | Raped | 1.46 | 24 |

The weighted log odds model supports the sentiment analysis findings that survivors displayed greater degrees of negative language in their interviews. Some of the negative words most specific to survivors' interviews pertain directly to negative life events, such as "trauma" and "PTSD". Increased reports of negative life experiences may explain part of the increased negative sentiment present in survivors' interviews. This is suggestive of a matrix of harm experienced by survivors, with multiple negative life-events being associated with assault. While the weighted log odds model detected words that systematically separated the interviews of survivors from those who did not report an assault experience, this model could not reveal why or how such words were used. To further investigate the words and sentiments specific to survivors' interviews, we qualitatively investigated how the words detected by the weighted log odds model appeared in the AVP interviews.

## Qualitative findings

Before qualitatively examining the words most specific to survivors' interviews, we clustered these words into three groups: assault-related language, trauma-related language, and non-assault-specific negative language. Tables 9–11 summarize word usage among survivors and those who did not report experiences of assault for these three groups.

Beginning with the term "raped," we examined how survivors, as opposed to those who did not report an assault experience, used this word. Of the 53 survivors who used the word "raped" in their interview, 42 of them used the word to describe a rape they experienced or to describe a personal experience of attempted sexual harm.

**Table 10. Use of trauma-related language by those who did and did not report an assault.**

|  |  | Average Use per Interview | Number of Interviews |
|---|---|---|---|
| Reported Assault |  |  |  |
|  | PTSD | 1.62 | 24 |
|  | Therapy | 3.59 | 74 |
|  | Trauma | 2.93 | 29 |
| Did not Report Assault |  |  |  |
|  | PTSD | 1.50 | <11 |
|  | Therapy | 2.42 | >70 |
|  | Trauma | 2.12 | >15 |

We coarsened the data and do not report the precise number of those who did not Report Assault to protect respondent confidentiality and to comply with AVP's disclosure avoidance policy

**Table 11. Use of word most specific to those who reported an assault.**

|  |  | Average Use per Interview | Number of Interviews |
|---|---|---|---|
| Reported Assault |  |  |  |
|  | Fucking | 8.56 | 34 |
| Did not report Assault |  |  |  |
|  | Fucking | 4.94 | 33 |

The single interview question targeting experiences of assault asked about "unwanted sexual contact" rather than labels, such as rape, sexual assault, or sexual abuse. Despite the intentionality behind the structure of this question, nearly 40% of the survivors who used the word "raped" utilized this language when responding to the "unwanted sexual contact" question. The remaining 60% used the word "rape" without being prompted by an interviewer, often in response to questions about major life events or their life story. Notably, a small fraction of survivors responded to the target question with versions of the following: "I've never been raped if that's what you mean. But. . .". These survivors went on to describe experiences that met an operational definition of sexual assault and even rape in some cases. For example, one interviewee described multiple experiences of unwanted sexual penetration but while disclosing these encounters said, "I've never been full-on raped though". This is consistent with research findings that indicate a hesitancy to label assault experiences as assault [8].

Of the 24 interviewees who did not report an assault experience and who used the word "raped," most used the term to refer to instances where friends, family, or acquaintances experienced unwanted sexual penetration. The discrepancy in usage of the term "raped" between survivors and those who did not report an assault experience came from survivors' personal disclosures of "being raped". Such disclosures constituted most of the instances of the word "raped" in the survivors' interviews.

The terms "abuse" and "abusive" were the second most distinguishing words among survivors divulging experiences of unwanted sexual contact. Survivors used these terms more than "raped," and used abuse-related language to describe firsthand experiences beyond those of sexual harm. Survivors did use the term "abuse" to refer to sexual assaults, but they also used this term to disclose instances of physical, verbal, mental, and emotional abuse. For example, one survivor reported, "I was mentally, physically, sexually abused as a child and that can mess with somebody". Another told an interviewer, "All of my marriages have been real abusive. My first husband was real horrible. . .. He would knock me out and [then] get real scared because I guess he thought he really hurt me or something". One survivor disclosed, "He was actually very emotionally abusive to me". While some survivors grouped experiences of sexual abuse with other abuses (often when referring to a single perpetrator of such abuses), others described sexual abuse and other abuse as distinct, mentioning one in a given section of the interview without mentioning the other. Survivors not only used the term "abuse" to refer to multiple forms of abuse, including physical, verbal, mental and sexual abuse, but frequently used the terms "abuse" and "abusive" in a first-person narrative, disclosing personal experiences of abuse. Again, this is suggestive of a matrix of harm experienced by survivors of sexual violence.

When those who did not report an assault experience used "abuse" and "abusive" in their interviews, such terms rarely referred to firsthand experiences of abuse. They used these terms in other contexts: "I don't believe in abuse;" "abuse the system;" "he was never abusive". Furthermore, those who did not report an assault experience frequently used abuse-related language to denote a circumstance that they had not experienced. One respondent who did not

report an assault experience said, "I never felt like any sexual abuse or anything like that". Similarly, another reported, "Never was abused or any of all that". Another, naming a type of abuse, asserted, "I never felt like any sexual abuse or anything like that". Those who did not report an assault experience not only used the terms "abuse" and "abusive" substantially less than survivors (as indicated by the weighted log odds model), but tended to employ these terms differently—and in ways that sentiment analysis cannot distinguish. While survivors often used abuse-related language to describe first-hand experiences of abuse, those who did not report an assault experience tended to use this language to indicate that they had not experienced harms. This suggests that our estimates of differences in sentiment are likely conservative, under-estimating the difference in negative sentiment between those who reported experiences of assault and those who did not.

Furthermore, several survivors connected their experiences of assault to repeated abuse. One survivor told interviewers, "I lost my virginity as a result of rape and I've been with a couple abusive relationships which could be physical or sexual [abuse]". Another survivor reported, "When I was [a young teenager], I was sexually abused. And then I got married and had my [child] when I was [in my twenties] and I was with a very abusive husband for. . .years".

After qualitatively examining assault-related language in the AVP interviews, we examined trauma-related language, specifically "PTSD," "therapy," and "trauma". Table 9 shows that twelve-times as many survivors as those who did not report the experience of assault used the word "PTSD". Survivors also used "PTSD" more frequently per interview than those who did not report experiences of assault. Additionally, nearly twice as many survivors used the word "trauma". While those who did not report experiences of assault used the term "therapy" slightly more than survivors, the survivors who included this term in their interviews used this word, on average, nearly four times per interview while those who did not report experiences of assault used it less than 2.5 times per interview.

Some survivors connected their experiences of trauma, therapy, and PTSD to sexual assault. One survivor discussed an experience of assault where she ended up in a close friend's place, thinking she was headed to a party. Instead, the man "busted out a lot of [drugs]" and "had kind of BDSM inclinations, but minus the consent". Describing the situation, she reported, "I was terrified. . .. I basically had to participate, and it was really traumatic to be honest. Yeah, that one took a lot of therapy to kind of get over". Reflecting on her experience of assault, this survivor wove trauma-related language into her story of sexual harm. She not only labeled the experience as "traumatic," but she reported attending "therapy" to cope with the trauma. Another survivor used similar language when reflecting on her understanding of her own experiences of sexual harm:

> In college . . . we were doing the unit on post-traumatic stress disorder, and we were talking mainly about deaths and victims of abuse, sexual abuse, and rape and incest. And I remember asking the teacher, something happened in my brain, and I raised my hand and asked the teacher, 'Is the incidence of PTSD different for victims of violent rape versus non-violent rape?' And the teacher goes, 'What's non-violent rape?' And then that was like when I realized that I had been raped.

This survivor used trauma-related language, namely "PTSD," when referencing sexual assault. While she did not specifically indicate that she experienced PTSD because of rape, she described how learning about the connection between PTSD and rape helped her acknowledge and label her own experience. These mentions of trauma-related language in the context of survivors' experiences of sexual assault represent a primary context where survivors' use of

trauma-related language departed from that of those who did not report an experience of assault.

Survivors also used trauma-related language more frequently outside of explicitly describing sexual assault. For example, survivors demonstrated greater willingness to use the term "PTSD," without necessarily connecting "PTSD" to sexual assault or to themselves. Several survivors referenced PTSD in contexts where they excluded information on sources or causes of PTSD. One survivor reported, "Well I had PTSD that was undiagnosed until the end of my senior year". Another told an interviewer, "I was in therapy and things like that and so they had started talking to me about me being PTSD". While these interviewees' experiences of sexual assault could be related to their personal reports of PTSD, we cannot provide evidence for a direct connection. In addition to firsthand reports of PTSD, some survivors used PTSD in a third-person context: "She really has that PTSD or something" or "He didn't do the best job with his own kids. Probably because of PTSD". Survivors in the AVP data used the term "PTSD" more than those who did not report an experience of assault, when discussing third party's experiences of trauma. This suggests that PTSD served as a structuring device for how it was that survivors understood and approached the world. It also suggests that assault experiences are part of a broader matrix of harm, where PTSD and trauma reflect those broader harms.

After qualitatively examining trauma-related language in the AVP interviews, we examined the term most specific to survivors' interviews: "fucking". Table 10 reveals that a similar number of survivors and those who did not report an experience of assault mentioned this word at least once in their interview. The survivors' usage rate per interview was nearly double that of those who did not report experiences of assault. Survivors averaged nearly nine mentions per interview, which reflected the highest usage rate of any word. Both survivors and those who did not report experiences of assault frequently employed "fucking" to emphasize negativity: "I felt like complete fucking shit;" "I was in a bad fucking place;" and "It's really fucking annoying". Some interviewees used this term in positive contexts, such as "I fucking love him," but these usages were relatively rare. A few survivors used this term in relation to their assault, such as calling their perpetrator "a total fucking creeper," but these usages were also rare. Overall, all interview subjects employed "fucking" in a range of negative contexts throughout their interviews. Regarding this term, the distinction between survivors and those who did not report experiences of assault emerged in the frequency of use: survivors used this negative term nearly twice as frequently.

Upon qualitatively examining responses about the negative words associated with assault experiences, we found that survivors reported experiences that reflect a broad matrix of harm. Which, taking from Beth Richie's concept, highlights "the various forms of violence women experience and the multiple contexts within which it simultaneously occurs" [16]. This qualitative evidence provides further support for our argument of greater negative sentiment within survivor interviews.

## Limitations

This study contains several limitations. The AVP data created specific complications. Given the negative connotations around sexual violence, interviewees may have underreported experiences of sexual assault. The sensitive nature of sex and sexual violence could have heightened this possibility. Those who elected to report assault experiences could have had more negative sentiments in general; selection into the sample, rather than assault experiences themselves, could be driving our findings. Further, not all respondents were asked the sexual assault question. With a nearly 200 question protocol, it is not surprising that questions were skipped. But

we also cannot rule out that selection into the sample of those asked the sexual assault question is driving some of our results.

The analysis itself contains limitations. The models were not able to account for certain features identified in existing literature as predictors of survivorship. Sexual orientation, previous victimization, and binge drinking could not be examined nor controlled for in the models as the AVP data did not provide robust information on these features. There were also limitations with the computational text analysis. We approached sentiment analysis by considering the text of each interview as a combination of individual words and the overall sentiment of each interview as the sum of the sentiment content of the words. The language lexicons employed, commonly used in computational text analysis, assign sentiment scores to individual words rather than phrases. As a result, these lexicons could misrepresent sentences where the sum of the sentiments of the individual words failed to equal the overall sentiment of the sentence. The lexicons themselves have not been validated for analyzing life histories. Our qualitative assessment sought to address this concern and provided some evidence that our estimates were likely conservative, but as we only examined some of the primary words driving the difference between groups, the concern remains. Finally, we cannot rule out priming through the informed consent process, and our robustness check that demonstrates negative sentiment before an assault question is even asked has limitations.

## Contributions

Studies estimate that nearly 50% of women and 25% of men in the United States experience sexual assault in their lifetimes. Such results often come from convenience samples or crime victimization surveys whose validity has been questioned [2]. This study largely confirms the prevalence rates reported in the literature, and does so using a national probability sample, providing important verification of previous work. It also suggests that there may be significant racial and ethnic differences in assault experiences. The findings on racial/ethnic differences in prevalence are more mixed within the literature on assault. The observed differences reported in this paper require further evaluation and would be worthwhile for future researchers to explore.

The major contribution of this work is to move beyond the question of prevalence and to explore the relationship between assault and people's whole lives. Through a novel exploration of the relationship between language/sentiment and assault this paper provides a new finding. The experiencing sexual assault is associated with negative sentiment that pervades survivors' descriptions of their whole lives. Those who report an experience of assault narrate aspects of their life that are not directly related to sexual assault in more negative ways than those who did not report an experience of assault. When we restricted the analysis to the portion of the interview before they were asked about assault, our results remained robust. This finding is novel. Previous research has typically studied the effect of sexual assault on the way survivors conceptualize and narrate their specific experience of assault. Our study contributes something that has not been observed before, namely that sexual assault survivors present their *entire lives* to others in more negative ways than those who did not experience assault. Across two lexicons and two sentiment scales, AVP survivors consistently expressed greater degrees of negative sentiment compared to those who did not report an experience of assault. When breaking down interviews into eight different emotions, survivors expressed greater degrees of all the negative emotions and lower degrees of the positive emotions. Survivors of assault narrate their life experiences with more anger, disgust, fear, and sadness, and less anticipation, joy, and trust than those who do not report assault experiences. We found that these differences were consistent for both men and women. Overall, this study provides evidence that

experiences of assault are part of a matrix of harm that pervades people's lives, interwoven with a range of negative experiences.

It is difficult to conceptualize what a 3-percentage point difference in negative sentiment means. To make this more interpretable, we conducted a test of comparative impact, examining interview sentiment across two other hardship variables: recent extreme financial hardship (proxied by receipt of a housing voucher in the last month) and the death of one or more parents. We found that a history of sexual assault influenced negative sentiment to a greater degree than the hardship of losing a parent. In contrast, the impact of recent financial hardship on negative sentiment exceeded that of lifetime sexual assault. Notably, many of the assaults reported were likely to have happened years and even decades before the interview; and still such assault experiences translated into only modestly lower increases in negative sentiment than experiences of receiving a housing voucher. This comparison suggests a sizable long-term impact of sexual assault on sentiment and language. This is a unique finding for the field that would benefit from further evaluation from multiple data sources.

While a small portion of previous literature has examined sentiments in response to hearing about sexual violence [26], this study marks the first exploration of sentiment of assault survivors and it does so from a national probability sample. It establishes a new zone of impact—sentiment—which work in psychology suggests may be meaningfully associated with life experiences. We evaluated that sentiment quantitatively, using a log-odds model to evaluate specific word usage, and qualitatively, contextualizing how such words were used within narratives themselves. Survivors more frequently used the words "raped," "abuse," "abusive," "PTSD," and "trauma" and they were more likely to do so in first-person contexts. Survivors reported more firsthand experiences with a range negative life events than those who did not report experiences of assault. Survivors frequently also employed several of these words, such as "PTSD" and "trauma," in third-person contexts. We found evidence that PTSD served as a "structuring device" for how survivors understood and approached the world. Survivors used negative language to describe the circumstances and behaviors of both themselves and others. While existing research investigates the scope of the impact of sexual assault on survivors [27], this paper suggests that this scope extends to items as fundamental as sentiment and negative language, permeating into the heart of how survivors describe their lives. This supports our concept of a "matrix of harm". We also provide evidence that our estimates may be conservative, given that those who did not experience assault were much more likely to use words like "abuse" when describing something they did *not* experience. Such a distinction is only discoverable through qualitative analysis (and not our quantitative sentiment analysis), showing the value of mixed methods approaches when conducting sentiment analyses.

We did not study the downstream behavioral and experiential effects of such life descriptions here. Future research should examine these sentiment effects in a causal framework. We do know that significant increases in negative emotions like anger, disgust, fear, and sadness, and decreases in positive emotions like anticipation, joy, and trust, are likely to come with negative consequences. Psychology literature has demonstrated the causal effects of such emotions on the quality of relationships, sleep, economic choices, politics, creativity, physical and mental health, and well-being [19]. As such, we suggest that the kind of sentiment analysis provided indicates the existence of a matrix of harm associated with assault experiences.

Finally, the strong association between sexual violence and negative sentiment suggests that a potential important determinant of life chances is consistently unmeasured. Such omitted variable bias exists within both quantitative and qualitative research instruments. This potential failure to fully understand important life outcomes will disproportionately impact our understanding of women, the LGBTQIA community, and those with a disability, all of whom experience significantly higher rates of sexual assault. Given its extremely high prevalence,

scholars may well be missing an explanation for major life outcomes, or misattributing effects because of our failure to observe this important part of a broader matrix of harm.

## Acknowledgments

This paper is based upon an undergraduate thesis written by McKenzie Caputo [28]. Thanks to Filiz Garip, David Grusky, Max Besbris, Betsy Paluck, and Brandon Stewart for comments on this project.

## Author Contributions

**Conceptualization:** MacKenzie Caputo, Max Fineman, Shamus Khan.

**Data curation:** MacKenzie Caputo, Max Fineman.

**Formal analysis:** MacKenzie Caputo, Max Fineman.

**Investigation:** MacKenzie Caputo.

**Methodology:** MacKenzie Caputo, Shamus Khan.

**Project administration:** MacKenzie Caputo, Shamus Khan.

**Supervision:** Max Fineman, Shamus Khan.

**Validation:** MacKenzie Caputo, Max Fineman, Shamus Khan.

**Visualization:** MacKenzie Caputo.

**Writing – original draft:** MacKenzie Caputo, Shamus Khan.

**Writing – review & editing:** Max Fineman, Shamus Khan.

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
