## [Decision Letter · Decision Letter 0]

11 Sep 2023

PONE-D-23-23689Sexual Assault Survivors narrative their whole lives in more negative ways PLOS ONE

Dear Dr. Khan,

Thank you for submitting your manuscript to PLOS ONE. After careful consideration, we feel that it has merit but does not fully meet PLOS ONE’s publication criteria as it currently stands. Therefore, we invite you to submit a revised version of the manuscript that addresses the points raised during the review process.

We look forward to receiving your revised manuscript.

Kind regards,

Rijen Shrestha, M.D.

Academic Editor

PLOS ONE

Journal Requirements:

4. Please ensure that you include a title page within your main document. You should list all authors and all affiliations as per our author instructions and clearly indicate the corresponding author.

5. Ethics statement does not appear in the manuscript file:

Please include your full ethics statement in the ‘Methods’ section of your manuscript file. In your statement, please include the full name of the IRB or ethics committee who approved or waived your study, as well as whether or not you obtained informed written or verbal consent. If consent was waived for your study, please include this information in your statement as well. 

6. Please include a copy of Table 4 and 6 which you refer to in your text on page 20 and 22.

Additional Editor Comments:

Please include a minor revision that includes a statement on “What this study adds to our current knowledge”.

Reviewers' comments:

Reviewer's Responses to Questions

**Comments to the Author**

1. Is the manuscript technically sound, and do the data support the conclusions?

Reviewer #1: Yes

Reviewer #2: Yes

2. Has the statistical analysis been performed appropriately and rigorously? 

Reviewer #1: N/A

Reviewer #2: Yes

3. Have the authors made all data underlying the findings in their manuscript fully available?

Reviewer #1: Yes

Reviewer #2: Yes

4. Is the manuscript presented in an intelligible fashion and written in standard English?

Reviewer #1: Yes

Reviewer #2: Yes

5. Review Comments to the Author

Reviewer #1: (1) The author of this paper, after using scientific instruments such as “using quantitative sentiment analysis”, arrives at the conclusion, that those who have experienced sexual assault narrate their lives more negatively than those who did not. Furthermore, the author says himself in his abstract, “Overall, this paper suggests that sexual violence is part of what we call, following Beth Richie, a “matrix of harm” that structures people’s lives.” Thus, the author is admitting that people have already said, what he has scientifically “discovered” in this paper.

(2)The author then goes on for a detailed definition/analysis/discussions of oft-repeated topics like “sexual violence” “matrix of harm”. “Impacts of negative statement” etc. The author also makes use of several interviews carried out under the “American Voices Project” From 2019 to 2021, and then describes a complex “Analytic Strategy” to come to the above conclusion.

(3) I think even an ordinary layman knows this already. Even from personal experiences, most of us know that any incident which is violent and brings harm to us, fills our life with negative emotions. Thus, I do not find anything new in this paper. If the author may resubmit his paper with just one additional paragraph entitled – “What this study adds to our current knowledge” it would make everybody understand properly what exactly is the new content in this paper. Importantly, in many journals, this column is an essential requirement.

(4) I am just wondering if this might be a case of “complexity bias”, where individuals have a tendency to overcomplicate simple things. But I will give the author a chance, and repeat my simple request. Add a final note – What exactly does this study add to our knowledge, and how does that “additional knowledge” help us in any manner.

Reviewer #2: The submitted manuscript hold its value in the field of forensic medicine. Very good presentation of data, results and other relevant facts. Limitations are also defined in the manuscript appropriately. Manuscript is worth publishing.

Thank you

6. PLOS authors have the option to publish the peer review history of their article (what does this mean?). If published, this will include your full peer review and any attached files.

Reviewer #1: No

Reviewer #2: No

---

## [Author Response · Author response to Decision Letter 0]

17 Nov 2023

A new section has been added as requested. It is entitled, “Contributions.” This section is after our limitations section and has been combined with our previous section entitled “conclusion.” It draws upon our previous conclusion but adds considerably more material as well as clarifying language about what the novel contributions of this paper are.

---

## [Decision Letter · Decision Letter 1]

10 Jan 2024

Sexual assault and the matrix of harm: Sexual assault survivors narrate their whole lives in more negative ways

PONE-D-23-23689R1

Dear Dr. Khan,

We’re pleased to inform you that your manuscript has been judged scientifically suitable for publication and will be formally accepted for publication once it meets all outstanding technical requirements.

Kind regards,

Rijen Shrestha, M.D.

Academic Editor

PLOS ONE

Additional Editor Comments (optional):

Reviewers' comments:

Reviewer's Responses to Questions

**Comments to the Author**

1. If the authors have adequately addressed your comments raised in a previous round of review and you feel that this manuscript is now acceptable for publication, you may indicate that here to bypass the “Comments to the Author” section, enter your conflict of interest statement in the “Confidential to Editor” section, and submit your "Accept" recommendation.

Reviewer #1: All comments have been addressed

Reviewer #3: (No Response)

2. Is the manuscript technically sound, and do the data support the conclusions?

Reviewer #1: Partly

Reviewer #3: Yes

3. Has the statistical analysis been performed appropriately and rigorously? 

Reviewer #1: I Don't Know

Reviewer #3: Yes

4. Have the authors made all data underlying the findings in their manuscript fully available?

Reviewer #1: Yes

Reviewer #3: Yes

5. Is the manuscript presented in an intelligible fashion and written in standard English?

Reviewer #1: Yes

Reviewer #3: Yes

6. Review Comments to the Author

Reviewer #1: (1) I received a book length document this time [184 pages] compared to just 53 page document that I received last time. Besides the revised paper, I could find a letter from a Princeton university professor of sociology. I am not sure, if this was deliberate or a genuine mistake. Anyway, what I could understand definitely is that this paper has been written by a sociologist and I am a simple forensic pathologist, with almost no education in sociology.

(2) In place of “What this study adds to our current knowledge” the author has decided to add “Contributions.” I can only evaluate this paper from a forensic angle, and from that angle, I still do not find anything new in it. But now I do understand, that this paper is written by a sociologist, and quite possibly, I am overlooking any sociological message inherent in this paper. This is attributable to my lack of knowledge of sociology.

(3)Purely from a forensic angle, it is of no/minimal value. I wish, I had the choice of passing over this paper to someone else with a knowledge in sociology. Since your system allows only “accept” or “reject” and some other choices in between, I must choose “reject”

Reviewer #3: The paper under review entitled “Sexual assault and the matrix of harm: Sexual assault survivors narrate their whole lives in more negative ways” is developed from the AVP data by the author to show the socially meaningful evidence of the harms associated with sexual assault. The paper shows that presents that those who have experienced sexual assault narrate their lives more negative manner, filled with anger, disgust, fear, and sadness, and less anticipation, joy, and trust. The sexual assault survivors have shown less joy in life and also less trust on others. The victim suffered lifelong as the results of the paper shows that the assaults that happened years and even decades earlier translated into only modestly lower increases in negative sentiment. It also reported that assault is part of a matrix of harm that pervades the lives of survivors. It further suggests that sexual violence is not randomly distributed across the population; its increased prevalence among women and LGBTQIA which indicates an unequal distribution of negative sentiment on a population level. The study provides evidence that experiences of assault are part of a matrix of harm that pervades people’s lives, interwoven with a range of negative sentiments.

Following are my comments on the paper:

1. The paper should be checked thoroughly for the language, editorial mistakes and journal guidelines. The readers may have difficulty in understanding due to ambiguity at some places.

2. It has also been found that if the grievances of such sexual assault survivors are not addressed properly through system, the negativity is even more among them.

3. Also, the time period should be taken more instead of only one month in case ‘recent extreme financial hardship’ and in case of ‘the death of one or more parents’ the time is not given.

The paper is good and presented in scientific manner and can be considered for publication.

7. PLOS authors have the option to publish the peer review history of their article (what does this mean?). If published, this will include your full peer review and any attached files.

Reviewer #1: No

Reviewer #3: **Yes: **RAJEEV KAMAL KUMAR

---

## [Editor Report · Acceptance letter]

7 May 2024

PONE-D-23-23689R1 

PLOS ONE

Dear Dr. Khan, 

I'm pleased to inform you that your manuscript has been deemed suitable for publication in PLOS ONE. Congratulations! Your manuscript is now being handed over to our production team.

Kind regards, 

on behalf of

Dr. Rijen Shrestha 

Academic Editor

PLOS ONE